# Object Detection Combining CNN and Adaptive Color Prior Features

**DOI:** 10.3390/s21082796

**Published:** 2021-04-15

**Authors:** Peng Gu, Xiaosong Lan, Shuxiao Li

**Affiliations:** 1School of Artificial Intelligence, University of Chinese Academy of Sciences, Beijing 100049, China; gupeng2018@ia.ac.cn; 2Institute of Automation, Chinese Academy of Sciences, Beijing 100190, China; lanxiaosong2012@ia.ac.cn

**Keywords:** convolutional neural network, color prior model, object detection

## Abstract

When compared with the traditional manual design method, the convolutional neural network has the advantages of strong expressive ability and it is insensitive to scale, light, and deformation, so it has become the mainstream method in the object detection field. In order to further improve the accuracy of existing object detection methods based on convolutional neural networks, this paper draws on the characteristics of the attention mechanism to model color priors. Firstly, it proposes a cognitive-driven color prior model to obtain the color prior features for the known types of target samples and the overall scene, respectively. Subsequently, the acquired color prior features and test image color features are adaptively weighted and competed to obtain prior-based saliency images. Finally, the obtained saliency images are treated as features maps and they are further fused with those extracted by the convolutional neural network to complete the subsequent object detection task. The proposed algorithm does not need training parameters, has strong generalization ability, and it is directly fused with convolutional neural network features at the feature extraction stage, thus has strong versatility. Experiments on the VOC2007 and VOC2012 benchmark data sets show that the utilization of cognitive-drive color priors can further improve the performance of existing object detection algorithms.

## 1. Introduction

Over the past few decades, the amount of visual image information has grown at an explosive rate. As a relatively accurate and vivid description of the objective world, visual image information is one of the mainstream forms for humans for understanding the world and receiving external information. Visual image information is difficult for computers to understand. How to process these image data and use the visual image information to improve people’s lives to the greatest extent is a very important subject, which is also the main problem in the field of computer vision [1]. Object detection is a very challenging research subject in the field of computer vision. Its main task is to output the bounding box position and classification confidence score of the target of interest in the test image [2]. Object detection algorithms can bring great convenience to our lives, such as the use of face detection algorithms to assist the camera’s autofocus, the use of medical image detection algorithms to assist doctors in accurately identifying disease features, and the use of object detection and tracking algorithms to realize the following flight of the UAV to the interested target, etc.

The main difficulties faced by the object detection algorithm are, as follows: (1) there are factors, such as uneven illumination, different postures, large scales, mutual occlusion, etc., which lead to problems, such as missed detection and false detection; (2) under the interference of complex background, the accuracy of the algorithm will drop sharply. In some environments with simple backgrounds and fixed scenes, traditional object detection algorithms can usually achieve satisfactory results. However, with the increasing demand of detection accuracy and speed in the fields of target tracking and automatic driving, traditional detection algorithms become unqualified. Therefore, how to detect targets in complex scenes and improve the detection speed is the focus of research in the field of object detection [3,4,5,6].

In recent years, deep learning technology has swept the field of artificial intelligence. Deep learning technology has made significant progress in computer vision [7], natural language processing [8], speech recognition [9], and other fields. Among them, the convolutional neural network technology relies on its advantages, such as scale invariance and translation invariance, and it has set off a storm of deep learning in the field of computer vision. Therefore, the convolutional neural network technology has become the mainstream algorithm in the current object detection field [10,11,12,13,14].

Visual attention is the cognitive process by which people obtain the most critical area information when observing natural scenes. Different parts of the human retina have different information processing capabilities. Therefore, this mechanism is very important for the efficient operation of the human visual system [15,16]. The human visual system will not consider equally for all areas in a natural scene in order to make reasonable use of the limited visual processing resources. Actually, the human visual system will selectively pay attention to the target area of interest, and ignore or suppress some less important background areas. There are two different methods of visual attention: one is bottom-up attention; the other is top-down driven attention that is based on specific tasks.

Being inspired by the top-down visual attention mechanism, we introduced the concept of color prior features [17]. Because different types of objects usually have different color features, this can help us to improve the ability to discern objects. Therefore, we propose a cognitive-driven color prior model for object detection. Two main contributions are included in our method. One aspect is that we propose an adaptive color prior model that comprehensively considers the fusion and competition among category pattern distribution (the category pattern distribution represents the probability of each pattern occurring in the target image block in labeled dataset for the class), scene pattern distribution (scene pattern distribution represents the probability of each pattern in all visual scene images), and test image pattern distribution (test image pattern distribution represents the probability of each pattern in the test image). It uses test image features to dynamically and adaptively adjust the memory prior to obtain more robust color prior information. The other aspect is that saliency images that are acquired by the proposed color prior model are regarded as feature maps and they are fused with those by convolutional neural networks, regardless of backbone network types, thus having strong versatility. The experiments show that the utilization of cognitive-drive color priors can further improve the performance of existing object detection algorithms.

The rest of the paper is arranged, as follows: Section 2 introduces related work, and Section 3 details the proposed methods. Section 4 analyzes the ablation experiments of the color prior model in Faster R-CNN and the contrast experiments in different types of object detection algorithms. Section 5 gives the main conclusions.

## 2. Related Work

This section mainly introduces related work from two aspects: object detection algorithm and saliency detection algorithm.

### 2.1. Object Detection Algorithm

According to different feature extraction methods, object detection algorithms can be roughly divided into traditional object detection algorithms and object detection algorithms that are based on deep neural networks.

Traditional object detection algorithms rely on carefully designed features to detect targets [18]. This type of method usually first designs a specific algorithm to screen out the most likely target areas, calculates the feature description of the candidate region according to the manually designed feature extraction rules, and then finally uses the pattern classification model to determine whether these candidate regions are real targets. The representative work of candidate region selection is sliding window and selective search algorithm [19]. The candidate region selection algorithm that is based on sliding window requires a large number of sizes and shapes of sliding windows, which will lead to huge computational redundancy. The selective search algorithm first divides the image into multiple small regions, and then further merges to obtain a larger candidate region, which improves the computational efficiency. Feature extraction is mainly designed for different tasks such as face detection and pedestrian detection. It includes the cognitive thinking wisdom of human experts. The most influential feature description methods include Scale-Invariant Feature Transform (SIFT) [20], Local Binary Pattern (LBP) [21], Histogram of Oriented Gradient (HOG) [22], etc. The main goal of pattern classification is to remove invalid candidate regions, which can be achieved by training effective classifiers. Commonly used algorithms are Support Vector Machines (SVM) [23], AdaBoost [24], etc. Although traditional object detection algorithms that are based on manual feature extraction have been developed to some extent since the end of the last century, there are still many shortcomings in practice. For example, HOG, SIFT, etc. are basically low-level features, such as contours and textures, which are less robust and cannot cope with object detection requirements in a diversified and complex environment.

The object detection algorithm that is based on deep neural network is a new type of method that has emerged in recent years. It can automatically form more abstract high-level features by combining low-level features from samples. These features have powerful expression and generalization capabilities. This is the current mainstream method of object detection [12,25,26]. In 2012, the AlexNet network [27] proposed by Alex Krizhevsky et al. achieved results far surpassing traditional object detection algorithms in the large-scale visual recognition challenge, which made deep neural network technology attract people’s attention in the field of image recognition and object detection. After several years of development, deep neural networks have been widely used in object detection tasks [28,29,30,31,32,33]. These algorithms are mainly divided into object detection algorithms that are based on candidate boxes and object detection algorithms based on regression. The object detection algorithm based on the candidate boxes is also called a two-stage type algorithm. It first extracts the region proposal, and then performs candidate boxes recognition and boxes regression. R-CNN series is the representative work [30,31,34]. R-CNN [34] uses a selective search algorithm to extract candidate frames, and it starts to use deep neural networks to extract features, and finally uses support vector machines to complete the classification of the target. Fast R-CNN [30] performs feature pooling for each candidate frame, and uses the softmax classifier to replace the support vector machine. It only needs to extract the features of the image once, which improves the training and reference speed. Faster R-CNN [31] uses neural networks to generate candidate boxes, instead of selective search algorithms, so that the entire object detection truly realizes end-to-end calculation. At the same time, region proposal, classification, and regression share convolution features, greatly improving the accuracy and computational efficiency of the algorithm. The object detection algorithm that is based on the regression idea is also called a single-stage type algorithm, which skips candidate box extraction stage and directly regards the object detection algorithm as a regression task, such as YOLO [32] and SSD [33]. For an input image, the regression-based object detection model directly outputs the categories and positions of all targets for each image block, and all of the detection steps are completed in a neural network. The advantage is that the algorithm is efficient, but the accuracy is usually slightly lower than that of the two-stage algorithm.

### 2.2. Saliency Detection Algorithm

In human cognitive science, different parts of the retina have different information processing capabilities. In order to make rational use of limited visual information processing resources, the human visual system usually selectively focuses on specific parts of the visual scene. This phenomenon is called the visual attention mechanism, which is the theoretical basis of saliency detection. The current saliency detection methods can be roughly divided into two types, one is a bottom-up saliency detection algorithm, which is data-driven, and the other is a top-down saliency detection algorithm, which is task-driven.

The bottom-up saliency detection algorithm directly extracts the underlying information of the image for detection, and it does not need to specify the type of target of interest or provide training samples in advance, so it has a wide range of adaptations. Itti and Koch et al. proposed a saliency detection algorithm earlier by learning from neuron structure and animal vision [35,36]; Xie et al. [37] proposed using the Bayesian model to construct saliency maps; and, Cheng et al. [38] proposed a saliency detection method based on global contrast.

The top-down saliency detection method is task-driven, which is, it is guided by human subjective consciousness or target tasks. It first learns the basic information of the target object at the training stage with supervision, and then uses the learned information to increase the significance of the expected region or the target of interest. Jing et al. [39] proposed the use of supervision to obtain prior information from the data set. Liu et al. [40] proposed using conditional random fields to generate a saliency map based on the extracted features, such as the center surround histogram, multi-scale contrast, and color space distribution. Zhang et al. [41] proposed not using a specific image set, but to obtain statistical information directly from the natural image set, and then build a saliency detection model that is based on the Bayesian method.

## 3. Proposed Method

### 3.1. Overall Structure

Figure 1 shows the overall flow of the algorithm. Figure 1a is the off-line memory stage. Based on training images, color priors are calculated through probability statistics and saved in the form of look-up tables for on-line mapping. Figure 1b is the on-line mapping stage. Adaptive color prior weights are first calculated through pattern competition and fusion. Subsequently, the acquired color prior saliency features of the test image are obtained through indexing the lookup table. Finally, the image pyramid technique is utilized to obtain the corresponding levels of color prior features $\left\{C4,C5,C6\right\}$, which are fused with deep neural network features $\left\{P4,P5,P6\right\}$ to perform the subsequent detection tasks. The ideas behind the two stages are detailed below.

(1) “Off-line memory” stage: drawing on the characteristics of human memory, the category pattern distribution of typical targets and the category-independent scene pattern distribution are separately established based on object detection databases and stored in a table. In order to facilitate memory and calculations, patterns are used to represent the local characteristics of the target (the patterns are essentially limited one-dimensional discrete numbers, such as grayscale, serialized colors, local binary patterns, etc., this article focuses on YUV color features). For a given object detection dataset, the category pattern distribution is established by aggregating the pattern frequencies in the target labeling boxes of the same category, and the scene pattern distribution is calculated by aggregating the pattern frequencies of all training images.

(2) “On-line mapping” stage: the object of interest in object detection usually only occupies a small part of the image, and it is “significant” relative to the test image. Therefore, only colors that are “significant” in the category pattern distribution and “rare” in both the scene pattern distribution and the test image pattern distribution should be enhanced. Thus, the saliency value of the color prior needs to be adaptively adjusted according to the test image. The saliency value of the color prior is obtained through the fusion and competition between the category pattern, the scene pattern, and the test image pattern, thereby establishing a dynamic adaptive color prior model that is represented by $\left\{pattern,saliency\right\}$. Borrowing the characteristics of human memory, the color saliency features of the test image are then obtained in the form of index lookup table [17], and are finally fused with the convolutional neural network features to work together on the object detection task.

### 3.2. “Off-Line Memory” Stage

This stage first introduces the representation method of the asymmetric color pattern image, and then gives the calculation process of the category pattern distribution and the scene pattern distribution, respectively, which can be saved offline.

#### 3.2.1. Asymmetric Color Pattern Image

In this paper, patterns are used to represent the local features of images, so that memory and mapping can be achieved through probability statistics and indexing, respectively. Let $I\in R^{w\times h\times 3}$ be the input image, where *w* and *h* are the image width and image height, respectively. We use the pattern operation Φ to transform the input image to a pattern image $B\in Z^{w\times h}$ (discrete integer space), which can be expressed as:(1)B=Φ(I)

Pattern operation Φ can be the serialization of color features, or texture feature operators, such as local binary pattern and ordinal features. In fact, when humans perform color matching, they pay more attention to the difference of color components, and allow a certain change in the brightness component to achieve illumination invariance to some extent. In order to express this characteristic, the RGB format image is converted to the YUV format image, and the asymmetric color pattern image is obtained. As we know, the YUV color model is closer to the human visual mechanism than RGB color model, thus it is more suitable for establishing the color prior model. In this paper, we quantize and splice the values of the three channels of Y, U, and V to obtain an asymmetric color pattern image. Quantizing the color channels can reduce the total number of patterns. In this paper, the brightness component Y is quantized to 16 levels, and the color components U and V are quantized to 32 levels, so as to allow certain brightness changes under the premise of better color distinctness. Thus, each pattern occupies 14 bits, and the value range is [0,16383]. Let $I_{Y},I_{U}$ and $I_{V}$ be the values of the Y, U, and V channels of the input image *I*, the calculation process of the asymmetric color pattern image can be expressed as:(2)$$\begin{array}{cc} \; & B_{c}=Con\left\{I_{Y}\gg 4,I_{U}\gg 3,I_{V}\gg 3\right\}\\ \; & =\left(I_{Y}\gg 4\right)\ll 10+\left(I_{U}\gg 3\right)\ll 5+\left(I_{V}\gg 3\right) \end{array}$$

Converting YUV images to pattern images is beneficial for memory and mapping. Memory can be simply done by summing up the pattern occurrence frequencies in target labeling images, from which color prior saliency for different types of targets can be learned and stored in advance. When recognizing the world, the memorized priors can be picked up by indexing, which is similar to human conditioned reflex.

#### 3.2.2. Category Pattern Distribution and Scene Pattern Distribution

There are many ways to express color features. This article adopts the form of pattern histogram. The pattern histogram expresses the probability distribution of different color patterns in typical category targets and scene images. It discards the spatial location information of color patterns in different images, thus it is translation and rotation invariant. For the object detection data set, each image may contain multiple targets of different categories, so we perform category pattern statistics that are based on the target block image $I_{o}$ in the target labeling box. Denote $I_{o}^{n,k}$ as the *k*-th target block image of the *n*-th category target, then its pattern probability distribution can be obtained by statistical histogram and normalization, which is recorded as:(3)$$\begin{array}{cc} \; & P^{n,k}=Normalize\left\{Hist\left\lfloor \Phi_{c}\left(I_{o}^{n,k}\right)\right\rfloor \right\}\\ \; & =\{p_{i}^{n,k},i=0,1,\cdots ,16383\} \end{array}$$

The category pattern distribution can be obtained by continuously “memorizing” the pattern probability distribution of all the target block images of the same category. For the training data set without time label, we take the mean value of all target block images of the same category as the final probability pattern distribution:(4)$$P^{n}=\left\{p_{i}^{n}=\frac{1}{N^{n}}\sum_{k=1}^{N^{n}}p_{i}^{n,k},i=0,1,\cdots ,16383\right\}$$
where $N^{n}$ is the total number of labeled samples of the *n*-th category.

There are always many different objects in natural scenes. The visual system usually weights key areas via a top-down manner based on experience or interest in order to make rational use of visual resources. This can be achieved through a visual competition mechanism. If the occurrence frequency of pattern *i* of the *k*-th image is higher than that of natural scenes or other types of targets, it means that the pattern *i* of the *k*-th image is “competitive” and it should be enhanced, otherwise it will be suppressed. Denote $P^{m}=\left\{p_{i}^{m},i=0,1,...,16383\right\}$ as the pattern distribution of the *m*-th training image; we approximate the scene pattern distribution $Q^{0}$ with the mean of all training images:(5)$$Q^{0}=\left\{q_{i}^{0}=\frac{1}{M}\sum_{k=1}^{M}p_{i}^{k},i=0,1,\cdots ,16383\right\}$$
where *M* is the total number of all training images.

### 3.3. “On-Line Mapping” Stage

This stage first establishes a dynamic adaptive color prior model through the fusion and competition between category pattern, scene pattern, and test image pattern. Subsequently, indexing and pooling operations are employed to generate different levels of color prior saliency features, which are finally combined with the convolutional neural network features to jointly achieve the object detection task.

#### 3.3.1. Dynamic Adaptive Color Prior Model

In the cognitive world, the human cerebral cortex reflexively obtains the prior information of typical objects to assist object detection. In this paper, the color space is converted into a pattern value, so that the prior saliency value of the color pattern of the corresponding category object can be retrieved in order to assist in identifying important objects. Specifically, the color prior model of the *n*-th category object can be expressed as:(6)$$W^{n}=\left\{w_{i}^{n},i=0,1,\cdots ,16383\right\},n\in \left[1,N\right]$$
where $w_{i}^{n}$ is the prior saliency value of the *i*-th color pattern of the *n*-th object class and *N* is the total number of object categories.

In the object detection scene, the object usually only occupies a small part of the image, and it is “significant” relative to the test image. Therefore, only colors that are “significant” in the category pattern distribution and “rare” in both the scene pattern distribution and the test image pattern distribution should be enhanced. That is to say, the saliency value of the color prior needs to be adaptively adjusted with the test image. For the test image *I*, we calculate its asymmetric color pattern image *B* according to formula (Equation 1), and then calculate the pattern distribution $Q^{t}$ of the test image, as below:(7)$$\begin{array}{c} Q^{t}=Normalize\left\{Hist\left\lfloor B\right\rfloor \right\}=\left\{q_{i}^{t},i=0,1,\cdots ,16383\right\} \end{array}$$

Subsequently, the pattern weight can be obtained by fusion and competition among the object pattern, scene pattern, and test image pattern:(8)$$W^{n}=\left\{w_{i}^{n}=\frac{p_{i}^{n}}{\eta \cdot q_{i}^{0}+\left(1-\eta \right)\cdot q_{i}^{t}},i=0,1,\cdots ,16383\right\},n=1,2,\cdots ,N$$
where η is the forgetting factor of the scene pattern. The forgetting factor is introduced, because, in addition to the scene pattern, it also needs to consider the influence of the test image on the pattern weight.

#### 3.3.2. Feature Map Generation

For the input image $I\in R^{w\times h\times 3}$, the pattern operation Φ is used to map it to the pattern image $B\in Z^{w\times h}$. For each pixel $x\in R^{2}$ in the pattern image, B(x)∈Z is the color pattern of the pixel. Taking B(x) as the index and reading the corresponding prior saliency values $\left\{w_{B(x)}^{n},n=1,2,\cdots ,N\right\}$ from the color prior model $W^{n}$ as its feature, we get *N* prior features for each pixel of the input image. Finally, the prior saliency values of all the pixels are organized in the form of multi-channel images to obtain priori saliency features $F\in R^{w\times h\times N}$ with *N* channels.

Subsequently, the obtained color prior saliency images are treated as features maps and they are fused with those extracted by the convolutional neural network to complete the subsequent object detection task. Taking Faster R-CNN as an example, the convolutional neural network uses ResNet-101, and it employs the feature pyramid network to extract multiple levels of features namely P2, P3, P4, P5, and P6. For the color prior model, we utilize the image pyramid technique to pool the saliency maps into multiple corresponding levels of saliency maps, namely C2, C3, C4, C5, and C6. We regard them as feature maps and fuse them with multiple hierarchical features extracted by the convolutional neural network together for the following object detection modules. In the detection task, the fused features are firstly input to the region proposal network to obtain regions of interests (ROIs). Subsequently, ROI pooling operation is utilized to extract regional features for each ROI. Finally, the acquired regional features are inputs to fully connected networks to output category probabilities and bounding box parameters.

## 4. Experiments

### 4.1. Datasets

This paper uses the cognitive-driven color prior model to improve the performance of the object detection algorithm. We verify the effectiveness of the proposed algorithm on the PASCAL VOC dataset. We use the AP (Average Precision) of each category object to measure the detection precision of the algorithm in different categories of objects, and then use the average precision of all categories to measure the advantages and disadvantages of the overall performance of the algorithm.

In the field of computer vision, PASCAL VOC is a set of standardized high-quality data sets, which are mainly used for tasks, such as object recognition, image segmentation, and object detection. PASCAL VOC marked a total of 20 categories of objects, namely: people, birds, cats, dogs, cattle, sheep, horses, chairs, bottles, potted plants, dining tables, TVs, sofas, bicycles, airplanes, buses, boats, cars, trains, and motorcycles. Figure 2 shows a typical example of the PASCAL VOC2007 data set. It contains 9963 images and a total of 24,640 labeled objects. It is composed of three parts, namely ghd train/val/test. The PASCAL VOC2012 data set is an upgraded version of PASCAL VOC2007. Figure 3 shows a typical example. The training set has 11,540 images and a total of 27,450 labeled objects. This paper uses the train/val parts of VOC2007 and VOC2012 as the training set, and the test part of VOC2007 as the test set to verify the proposed algorithm.

### 4.2. Implementation Details

The programming software for algorithm verification is the VsCode integrated development environment with Ubuntu 18.04 operating system and Cuda environment. We use an Intel i9-9920X CPU, 4 NVIDIA GeForce RTX2080TL GPUs and 96G DDR3 memories. The deep learning framework is Pytorch.

We use Faster R-CNN+FPN [42] to undertake ablation experiments on the PASCAL VOC data set, where the backbone network adopts ResNet-101 with pre-training weights on the ImageNet data set. The training period is set to 12, the batch size to 1, the initial learning rate to 0.001, and other parameters are consistent with the open source project [31]. The color prior model involves an important parameter η. In order to make the color prior model achieve the best results, we conduct a qualitative analysis on the parameter η, and finally take η = 0.9 for ablation experiments and comparison experiments.

We select four classic target detection frameworks for comparison experiments, including SSD [33], RetinaNet [43], Cascade-R-CNN [44], and Libra-R-CNN [45], in order to verify the generality of the color prior model. The implementation of these algorithms mainly refers to the mmdetection library. Mmdetection is an open source library that is based on Pytorch and it currently supports many mainstream object detection models. Among them, Libra-R-CNN [45] selects ResNet-50 [7] as the backbone network; SSD [33] sets the input size to 300 and selects VGG16 [46] as the backbone network; Cascade-R-CNN [44] selects ResNet-50 [7] as the backbone network; and, RetinaNet [43] selects ResNet-50 [7] as the backbone network.

### 4.3. Overall Performance Verification of the Algorithm

We select Faster R-CNN+FPN as the baseline and compare the AP of the object detection network with/without color priors on the VOC datasets in order to verify the effectiveness of the color prior model. Faster R-CNN selects ResNet-101 to extract the basic features of the input image, and it uses FPN to construct high-level semantic features at various scales. FPN adopts a top-down hierarchical structure with side links. The advantage is that it can make reasonable use of the inherent multi-scale and hierarchical structure of deep convolutional networks, so that features at different scales have strong semantic information. Through CNN+FPN, the test image obtains multiple feature layers, namely $\left\{P2,P3,P4,P5,P6\right\}$, and it generates saliency maps through the color prior model. The acquired saliency maps are then pooled into the corresponding multiple hierarchical features namely $\left\{C2,C3,C4,C5,C6\right\}$ through the image pyramid technique (the feature maps are illustrated in Figure 4). The two sets of features are spliced and input into the detection head RPN for the subsequent detection modules.

Table 1 shows the performance comparison of the object detection algorithm with/ without the color prior model. On the VOC2007 data set and VOC07+12 data set, the mAP of the object detection network combined with the color prior model surpasses the baselines by 1.1% and 1.0%, respectively. It shows that the color prior model is effective and it can improve the performance of the object detection algorithm on different data sets. Table 2 shows the speed comparison of whether to use the color prior model. We implement the color prior model in Python language in order to quickly verify the experimental effect. Because pixel-level image operations in Python are inefficient, the speed of current version is slow. We will reimplement it in C language in future work to improve the efficiency.

Figure 5 shows the result images of the benchmark algorithm and the proposed algorithm in a typical experiment of object detection. It can be seen from the figure that the benchmark algorithm has an error detection (left column in Figure 5), class error (middle column in Figure 5), and missed detection (right column in Figure 5). However, the object to be detected still has certain color characteristics, and the correct detection result can be obtained by combining the color prior model, which proves that the cognitive-driven color prior model can enhance the performance of the target detection algorithm.

### 4.4. Ablation Study

We conduct ablation experiments according to whether the following design ideas are included:YUV asymmetric color pattern: Another option is to use the RGB color pattern.Whether to use forgetting factor η.Fusion methods of color features and network features: a variety of feature fusion methods are designed and compared, and the experiments are performed to verify that the best feature fusion method is selected.

Table 3 shows the performance comparison of object detection algorithms using YUV asymmetric color pattern and RGB color pattern. The RGB color pattern is formed by quantizing the R, G, and B components into 32 levels and splicing them, thus each pattern occupies 16 bits. Meanwhile, the YUV asymmetric color pattern that is proposed in this article only occupies 14 bits, which reduces the storage capacity of the color prior model and increases the calculation speed. It can be seen from the table that the mAP of using the YUV asymmetric color pattern has advantages in both the VOC2007 data set and the VOC07+12 data set, indicating that it is more effective than the RGB color pattern.

In order to obtain more robust color prior information, we introduce the forgetting factor η, using test image features to dynamically and adaptively adjust the memory prior. Table 4 shows the performance comparison of object detection algorithms with/without the forgetting factor. From the table it can be seen that the color prior model using the forgetting factor is better than the case where the forgetting factor is not used on the VOC2007 data set and the VOC07+12 data set, which proves that the forgetting factor is useful for the color prior model.

In the feature fusion stage, we compared various fusion strategies, as shown in Table 5, where C represents the color prior features, P represents the neural network features, C_P represents splicing C and P, PH represents multiplying P by the channel mean value of C pixel by pixel as spatial saliency weighting, PH_P represents splicing PH and P, and PH+P represents the addition of PH and P. Through comparative experiments, it can be seen that the feature fusion method of C_P has the best effect. The PH_P and PH+P fusion methods can also improve the performance of the detection network, but the effect is not as good as the C_P method. The PH method reduces the performance of the object detection network. The reason may be that the spatial saliency weighting inhibits part of the effective feature expression of the neural network. We will continue to pay attention to the research in this area in future experiments.

### 4.5. Comparative Study

We select four classic object detection networks for grouping experiments in order to verify the universality of the color prior model. Table 6 shows the comparative experimental results on VOC2007 and VOC07+12 data sets. It can be seen from the table that: (1) color prior model can be used to improve the accuracy for all the above-mentioned object detection networks, which proves that the proposed method is universal to different object detection networks. (2) The object detection networks utilizing the color prior model obtained improved performances when compared with those of the original frameworks on both the VOC2007 data set and VOC07+12 data set. This proves that the color prior model has strong versatility under different data sets.

## 5. Conclusions

This paper proposes an object detection method utilizing the color prior model. Specifically, we first learn from the visual attention mechanism to calculate the scene pattern distribution and category pattern distributions from annotated datasets, and save them in the form of tables off-line. For the on-line phase, the scene pattern distribution, category pattern distribution, and test image pattern distribution are competed and fused to generate adaptive color pattern weights, based on which color prior features can be efficiently obtained through indexing. Finally, the acquired color prior features are fused with CNN features for the subsequent object detection modules. The proposed color prior model is cognitively driven and it has no training parameters, so it has strong generalization ability. The experiments show that color priors can effectively improve the performance of object detection networks with different structures. In future work, we will investigate the effects of other color patterns, explore the self adaptability method of the forgetting factor, and study the fusion strategy between the color prior model and the object detection network at different stages.

## Figures and Tables

**Figure 1 sensors-21-02796-f001:**
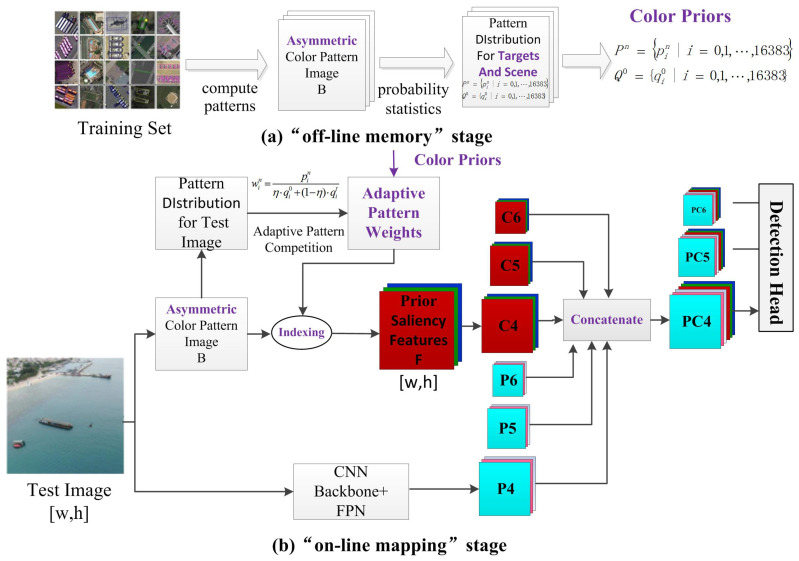
Overall architecture of object detection method combined with color prior model.

**Figure 2 sensors-21-02796-f002:**
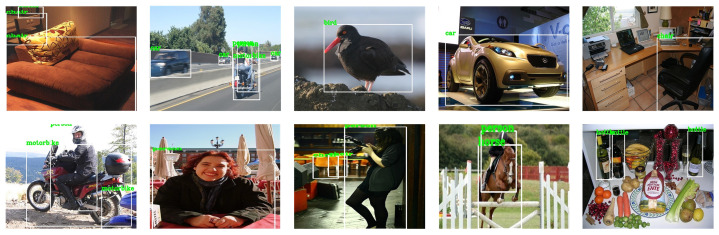
PASCAL VOC2007 datasets.

**Figure 3 sensors-21-02796-f003:**
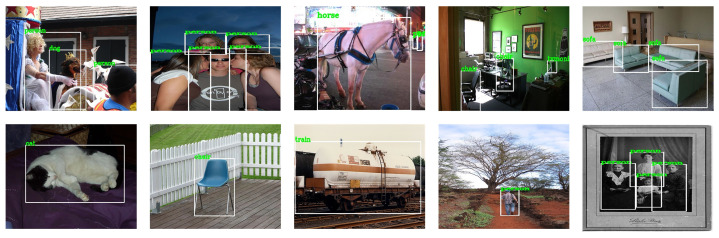
PASCAL VOC2012 datasets.

**Figure 4 sensors-21-02796-f004:**
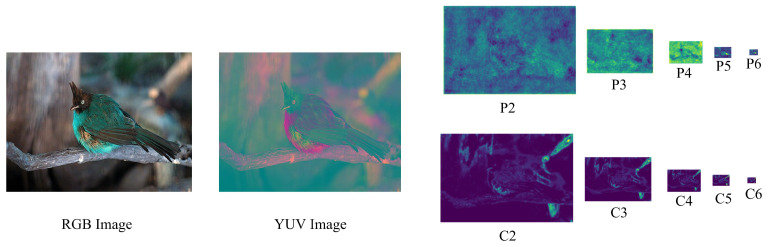
Test image and multi-level features.

**Figure 5 sensors-21-02796-f005:**
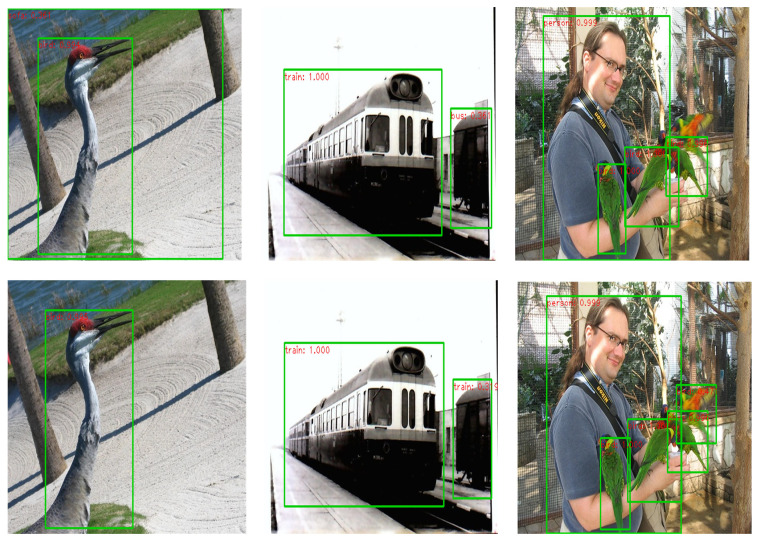
The typical result images of target detection, the first row is from the benchmark algorithm, and the second row is from the proposed algorithm.

**Table 1 sensors-21-02796-t001:** Performance comparison of whether to use the color prior model.

	Backbone	Datasets	ColorPriors	mAP
Faster R-CNN+FPN	ResNet-101	VOC2007	×	0.756
Faster R-CNN+FPN	ResNet-101	VOC2007	✓	0.764
Faster R-CNN+FPN	ResNet-101	VOC07+12	×	0.803
Faster R-CNN+FPN	ResNet-101	VOC07+12	✓	0.811

**Table 2 sensors-21-02796-t002:** Speed comparison of whether to use the color prior model (datasets:VOC2007).

	Backbone	ColorPriors	Train Speed/Epoch	Test Speed/Epoch
Faster R-CNN+FPN	ResNet-101	×	0.5 h	0.08 h
Faster R-CNN+FPN	ResNet-101	✓	0.66 h	0.16 h

**Table 3 sensors-21-02796-t003:** Performance comparison of object detection algorithms using different color spaces.

	Backbone	Datasets	ColorPriors	Color Pattern	mAP
Faster R-CNN+FPN	ResNet-101	VOC2007	✓	RGB	0.762
Faster R-CNN+FPN	ResNet-101	VOC2007	✓	YUV	0.764
Faster R-CNN+FPN	ResNet-101	VOC07+12	✓	RGB	0.808
Faster R-CNN+FPN	ResNet-101	VOC07+12	✓	YUV	0.811

**Table 4 sensors-21-02796-t004:** Performance comparison of object detection algorithms with/without forgetting factor.

	Backbone	Datasets	ColorPriors	η	mAP
Faster R-CNN+FPN	ResNet-101	VOC2007	✓	×	0.760
Faster R-CNN+FPN	ResNet-101	VOC2007	✓	✓	0.764
Faster R-CNN+FPN	ResNet-101	VOC07+12	✓	×	0.807
Faster R-CNN+FPN	ResNet-101	VOC07+12	✓	✓	0.811

**Table 5 sensors-21-02796-t005:** Performance comparison of object detection algorithms using different feature fusion strategies.

	Backbone	Datasets	ColorPriors	Fusion Strategy	mAP
Faster R-CNN+FPN	ResNet-101	VOC2007	✓	C_P	0.764
Faster R-CNN+FPN	ResNet-101	VOC2007	✓	PH	0.737
Faster R-CNN+FPN	ResNet-101	VOC2007	✓	PH_P	0.759
Faster R-CNN+FPN	ResNet-101	VOC2007	✓	PH+P	0.759
Faster R-CNN+FPN	ResNet-101	VOC07+12	✓	C_P	0.811
Faster R-CNN+FPN	ResNet-101	VOC07+12	✓	PH	0.779
Faster R-CNN+FPN	ResNet-101	VOC07+12	✓	PH_P	0.805
Faster R-CNN+FPN	ResNet-101	VOC07+12	✓	PH+P	0.804

**Table 6 sensors-21-02796-t006:** Comparative experiments of different target detection networks.

	Backbone	ColorPriors	mAP Using VOC07	mAP Using VOC07+12
Cascade R-CNN	ResNet-50	×	0.726	0.781
	ResNet-50	✓	0.732	0.788
SSD300	VGG16	×	0.707	0.775
	VGG16	✓	0.712	0.782
Libra R-CNN	ResNet-50	×	0.743	0.808
	ResNet-50	✓	0.748	0.813
RetinaNet	ResNet-50	×	0.712	0.793
	ResNet-50	✓	0.717	0.797

## Data Availability

Not applicable.

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
