# Peer review of "Object Detection Combining CNN and Adaptive Color Prior Features"

_sensors, 2021, doi:10.3390/s21082796_

Round 1

Reviewer 1 Report

In this paper authors present a fusion method of the Faster R-CNN+FPN and the color prior features, showing the combination of the color prior features with the CNN-based object detection scheme improves the mAP.

The idea that fuses color prior features to the feature maps of CNN-based scheme, especially the Faster R-CNN+FPN, may be novelty. In this sense, the paper can be published after some modifications.

I recommend the following modifications to make a better comprehension of the paper.

  1. The title of the paper does not represent the content of the paper, because authors show the improvement of accuracy combining the CNN-based scheme and color prior features. In my opinion, the term “CNN-based object detection” or “R-CNN” must be included in the title.
  2. In the proposed scheme, authors do not explain well about “Detection Head” after the fusion (Fig. 1). How to use the fused data to detect object?
  3. The fusion of the color prior features improves at most 1% of mAP. This improvement may be important; however, the fusion also increases the computational cost. Please add some information about the computational cost in the test stage as well as in the training stage.
  4. The proposed scheme uses three patterns: object pattern, scene pattern and test image pattern. I cannot understand well a role or meaning of each pattern. Please add some intuitive explanation about these patterns.
  5. The tables 5 and 6 can be combined adding one column for “mAP using VOC07+12”.
  6. The section of conclusions is very reduced. Authors must add more information of the paper in this section.

Author Response

Reply to reviewer 1:

Thank you for your review

1   The title of the paper does not represent the content of the paper, because authors show the improvement of accuracy combining the CNN-based scheme and color prior features. In my opinion, the term “CNN-based object detection” or “R-CNN” must be included in the title.

Reply:

The title of the paper is revised as“Object detection combining CNN and adaptive color prior features”

2   In the proposed scheme, authors do not explain well about “Detection Head” after the fusion (Fig. 1). How to use the fused data to detect object?

Reply:

 In the detection task, the fused features are firstly input to the region proposal network to get regions of interests(ROIs). Then, ROI pooling operation is utilized to extract regional features for each ROI. Finally, the acquired regional features are inputs to fully connected networks to output category probabilities and bounding box parameters.

3   The fusion of the color prior features improves at most 1% of mAP. This improvement may be important; however, the fusion also increases the computational cost. Please add some information about the computational cost in the test stage as well as in the training stage.

Reply:

In order to quickly verify the experimental effect, we implement the color prior model in Python language. Since pixel-level image operations in Python are inefficient, the speed of current version is slow. We will reimplement it in C language in future work to improve

the efficiency.

4  The proposed scheme uses three patterns: object pattern, scene pattern and test image pattern. I cannot understand well a role or meaning of each pattern. Please add some intuitive explanation about these patterns.

Reply:

Category pattern distribution represents the probability of each pattern occurring in the target image block in labeled dataset for the class.

Scene pattern distribution represents the probability of each pattern in all visual scene images.

Test image pattern distribution represents the probability of each pattern in the test image.

5  The tables 5 and 6 can be combined adding one column for “mAP using VOC07+12”.

Reply:

Tables 5 and 6 have been combined.

6  The section of conclusions is very reduced. Authors must add more information of the paper in this section.

Reply:

   We supplement the conclusion.

Reviewer 2 Report

1. It would be nice to see the discussion on the speed of the proposed method.
2. The explanation on the forgetting factor can be more detailed. Please explain how it is selected and whether it could be self-adpative.
3. Is there other color pattern could be applied? How the color prior model be supportive to color patterns? Discussion on this could be interesting to the readers.

Author Response

Reply to reviewer 2:

Thank you for your review

1  It would be nice to see the discussion on the speed of the proposed method.

Reply:

In order to quickly verify the experimental effect, we implement the color prior model in Python language. Since pixel-level image operations in Python are inefficient, the speed of current version is slow. We will reimplement it in C language in future work to improve

the efficiency.

2  The explanation on the forgetting factor can be more detailed. Please explain how it is selected and whether it could be self-adpative.

Reply:

The forgetting factor is introduced because in addition to the scene pattern, it also needs to consider the influence of the test image on the pattern weight. In future work, we will explore the self adaptability method of  the forgetting factor.

3  Is there other color pattern could be applied? How the color prior model be supportive to color patterns? Discussion on this could be interesting to the readers.

Reply:

As we know, YUV color model is closer to the human visual mechanism than RGB color model, thus is more suitable for establishing the color prior model. In future work, we will investigate the effects of other color patterns.

Reviewer 3 Report

This manuscript combines the color prior feature images with CNN features for object detection. the paper is overall clearly stated with some problems to be specified,
1) the abstract need to be abbreviated. especially the previous 4 sentences.
2) for the introduction part, more references and descriptions are needed for the the introduction to the color prior features.
3) for the abstract part, it is stated color prior features and real-time color features are weighted to obtain prior based saliency images. what is real-time color features, in the method part, the paper uses online mapping and offline memory stage, it is recommended to be revised.
4) why does this paper focus on YUV color model instead of other color models?

Author Response

Reply to reviewer 3:

Thank you for your review

1) the abstract need to be abbreviated. especially the previous 4 sentences.

Reply:

The first four sentences in the abstract have been abbreviated

2) for the introduction part, more references and descriptions are needed for the the introduction to the color prior features.

Reply:

Inspired by the top-down visual attention mechanism, we introduced the concept of color prior features. Since different types of objects usually have different color features, this can help us improve the ability to discern objects. Therefore, we propose a cognitive-driven color prior model for object detection.

3) for the abstract part, it is stated color prior features and real-time color features are weighted to obtain prior based saliency images. what is real-time color features, in the method part, the paper uses online mapping and offline memory stage, it is recommended to be revised.

Reply:

Real time color features represent test images pattern distribution.

4) why does this paper focus on YUV color model instead of other color models?

Reply:

As we know, YUV color model is closer to the human visual mechanism than RGB color model, thus is more suitable for establishing the color prior model. In future work, we will investigate the effects of other color patterns.

Round 2

Reviewer 1 Report

Authors attended correctly all suggestions provided by me.  

Additional suggestion about the revised version:

The position of table 6 must be changed, which must be before or after the conclusions.  

Author Response

Reply:

Thank you very much for your review and suggestions. Table 6 has been placed before the conclusions.

Reviewer 3 Report

my questions have been answered, but not quite satisfying.

Author Response

Reply:

Thank you very much for your review and suggestions. The basic principles and design ideas of the paper will be further improved in future work.

This manuscript is a resubmission of an earlier submission. The following is a list of the peer review reports and author responses from that submission.

Round 1

Reviewer 1 Report

-please add photo of measurement, if any?;;;
-please add block diagram of the proposed research step by step ;;; what is the result of paper?;;;
-Figure 1 it is good idea to describe or explain what is C4, C5, C6, etc.
-please add block diagram of the proposed method;;;
-please add photo/photos of application of the proposed research ;;;;
-please add sentences about future analysis;;;
-references should be 2018-2021 Web of Science about 50% or more ;;
-Please compare with other methods, justify. Advantages or Disadvantages;;;
for example with thermal analysis

1) Fault diagnosis of electric impact drills using thermal imaging, Measurement, Volume 171, 2021,
https://doi.org/10.1016/j.measurement.2020.108815

-Conclusion: point out what are you done;;;;
-please add relation to Electronics